# Repetitive negative thinking mediates the relationship between self-esteem and burnout in an ecological momentary assessment study
Malin Brueckmann [1,2] ✉, Justin Hachenberger [1,2], Elke Wild [1] & Sakari Lemola [1]

Low self-esteem and repetitive negative thinking are associated with higher burnout risk among university students at the between-person level. However, there is increasing evidence that associations identified in between-person analyses do not always reflect processes occurring within individuals. Therefore, we conducted a four-week ecological momentary assessment study with $N = 96$ students during an examination period. Results showed that higher self-esteem was followed by feeling less burnt out on a within-day and day-to-day level. Also, higher self-esteem was followed by lower repetitive negative thinking (i.e., rumination on the within-day level and pre-sleep worry on the day-to-day level), which in turn was followed by feeling less burnt out. Mediation analyses showed that a substantial proportion of the associations between self-esteem and feeling burnt out was mediated by repetitive negative thinking at both the within- and between-person level. In addition, we also found evidence of a reverse temporal sequence. Higher levels of burnout were followed by an increase in repetitive negative thinking, which in turn led to lower self-esteem. Finally, self-esteem instability partially moderated the associations of self-esteem and subsequent pre-sleep worry and burnout at the within-person but not between-person level. These findings imply that there may be a bidirectional relationship between self-esteem, repetitive negative thinking, and burnout, indicating a possible vicious cycle that could perpetuate psychological distress. Future studies should examine these dynamics more closely to better understand their causal interplay and implications for intervention.

Academic burnout is highly prevalent among university students (prevalence up to 28% in the German population[1]) and is associated with poorer academic performance[2], increased dropout rates[3,4], and impaired long-term well-being[5]. Initially explored in the workplace, research on burnout has since extended to academia, highlighting clear parallels[6]. Burnout is characterised by exhaustion, disengagement and reduced self-efficacy[7,8]. Longitudinal empirical studies suggest that exhaustion is the primary symptom of burnout. In the academic context, sustained study-related stress can lead students to feel emotionally drained and perceive their studies as burdensome. As a coping response, students may begin to disengage from their studies, which, over time, can undermine their sense of competence and academic efficacy[9–11]. Accordingly, exhaustion is considered the initial and central symptom of academic burnout in this study.

Study demands in the form of high workload and time pressure are a key risk factor for the development of burnout[12,13] and tend to peak in the period before upcoming exams. Furthermore, for many students, exam periods are particularly stressful[14,15], as the intense focus on measurable outcomes (such as grades) increases feelings of inadequacy and fear of failure[16,17].

The emotion regulation strategies that students use during and after stressful periods play a crucial role in recovery from stress. Repetitive negative thinking—such as worrying and rumination—acts as a maladaptive self-regulatory process that can hinder the reduction of psychological distress[18–21]. When students continue to think about the demands of their studies in their leisure time and thus remain attached to their studies, emotional exhaustion (the key symptom of burnout[22]) is more likely to occur.

[1]Department of Psychology, Bielefeld University, Bielefeld, Germany. [2]These authors contributed equally: Malin Brueckmann, Justin Hachenberger.
✉e-mail: malin.brueckmann@uni-bielefeld.de

In particular, students with low self-esteem have been suggested to be more likely to engage in repetitive negative (self-referential) thinking as they tend to focus on personal failures and shortcomings, especially in response to stressful experiences[23]. Theoretically and empirically, low self-esteem is considered a key vulnerability factor for repetitive negative thinking (e.g. rumination[24–26]) and ill-being (e.g. burnout[27]).

While the individual relationships between self-esteem, repetitive negative thinking, and burnout have been examined, to our knowledge, neither a joint consideration of all three constructs nor a specific examination of within-person and between-person effects has been undertaken. Given the critical consequences of academic burnout, understanding the underlying mechanisms (both within and between individuals) is essential for developing strategies to reduce distress and improve both well-being and academic success.

Global self-esteem refers to an individual's subjective evaluation of themselves, reflecting their overall attitude towards their own person[28]. The level of self-esteem indicates how much individuals value and like themselves, ranging from high and positive to low and negative. Low self-esteem is associated with increased mental health problems[29], including anxiety, depression[30] and burnout[27].

According to the Conservation of Resources Theory[31], which states that stress arises from a perceived threat to resources, self-esteem is considered a key personal resource that buffers stress[32]. High self-esteem serves as a protective factor, enabling individuals to cope more effectively with high demands, such as those encountered during exam periods. Consistent with this, higher self-esteem has been shown to result in a lower physiological response to challenging or achievement-related situations[33,34]. In contrast, individuals with lower self-esteem exhibit stronger cortisol responses to stress and tend to perform worse under pressure[35]. The results of an experimental neuroimaging study further suggest, that individuals with lower self-esteem might have to 'put more effort in cognitive performance control, stress and emotion regulation and furthermore, think about themselves and relive previous stressful experiences.' (p. 6)[34]. These increased stress responses and regulatory demands may deplete cognitive and emotional resources more quickly, thereby contributing to (emotional) exhaustion.

Thinking further, (emotional) exhaustion, followed by disengagement and reduced self-efficacy[9] may impair students' ability to achieve their goals (e.g. due to lower performance[36]). The findings of an experimental study show that academic failure drains students' state self-esteem[37], leading to a vicious cycle of low (state) self-esteem and burnout symptoms.

Traditionally, self-esteem has been conceptualised as a relatively stable trait[23,38] and regarded as a key vulnerability factor for psychological distress (e.g. burnout[39,40]). This perspective is supported by developmental considerations: While self-esteem develops from early childhood onwards, burnout is a later-emerging, context-specific phenomenon that arises in response to chronic role-related stress[8,22], especially when important resources are lacking. Consistent with this, empirical research shows that burnout symptoms increase during lower secondary education, and are closely associated with structured performance environments such as schools, universities, and workplaces[12,41]. Based on this reasoning, we conceptualise self-esteem as a precursor to burnout development.

At the same time, self-esteem also exhibits meaningful short-term variability[42]. Using a state-based measure of self-esteem in our study allows us to examine its predictive role and to test the reverse direction, i.e. whether daily experiences of stress and exhaustion predict subsequent decreases in self-esteem. Overall, while our study follows the theoretical tradition of modelling self-esteem as a predictor of burnout symptoms, it also acknowledges the potential for bidirectional effects by testing for the possibility that burnout-related states may, in turn, erode students' self-esteem.

Repetitive negative thinking refers to persistent, difficult-to-control thoughts about one's feelings and problems[43]. It is an umbrella term for both future-oriented worry and past-oriented rumination[44,45], which, although often studied separately[46], share common features, are highly intercorrelated[47,48], and contribute to psychopathology and somatic health complaints[49].

Research, for example, a five-wave longitudinal study[24], indicates that low self-esteem contributes to increased repetitive negative thinking, which can be explained by several mechanisms[50]: First, individuals with low self-esteem often hide their feelings, failures, and struggles from others[51], but suppressing negative emotions can further increase repetitive negative thinking[52]. Second, from an information processing perspective, individuals characterised by low self-esteem tend to bias their attention towards negative emotion and information[53], leading to repetitive negative thinking. Third, as many students with low self-esteem procrastinate during exam preparation[54], feel rather unprepared, and worry about how poor academic performance might affect their record[14,17,55], their basic need for competence may be threatened. Perceived threats to basic psychological needs, such as competence[56], not only trigger rumination but also contribute to its persistence over time[52]. In summary, there is ample evidence that lower self-esteem contributes to a higher frequency of past-oriented rumination and future-oriented worry.

Perseverative negative thoughts, including worrying and ruminating, e.g. about failing, may interfere with restorative resting episodes (e.g. reduced sleep quality[57]), which may drain energy levels. Consistent with the perseverative cognition hypothesis[44], both (past-oriented) rumination and (future-oriented) worry activate cognitive, emotional and physiological processes[49]. As 'repetitive' or 'perseverative' negative thinking means that the duration of the cognitive representation of stressors extends well beyond their occurrence, emotional and physiological activation persists during the time off study, ultimately reducing the time available for recovery. According to the Conservation of Resources Theory[31,58], which suggests that stress arises when there is a threat of losing resources (such as time and energy), a lack of recovery experiences is associated with several negative outcomes, including sleep problems[59,60] and burnout[61]. Therefore, theoretical and empirical evidence suggest that higher frequencies of rumination and pre-sleep worry contribute to academic burnout.

Similarly, experiencing burnout symptoms may lead to repetitive negative thinking, which can result in lower (state) self-esteem. In an academic context, students experiencing burnout symptoms, particularly in terms of (emotional) exhaustion and reduced self-efficacy, may ruminate more frequently about past failures or worry about future academic demands. Such repetitive negative thinking can reinforce negative self-perceptions and undermine momentary self-esteem. Longitudinal studies support this, showing bidirectional associations of repetitive negative thinking with both self-esteem[25] and burnout symptoms (e.g. exhaustion[62]). In summary, both theoretical considerations and empirical evidence suggest that repetitive negative thinking may be the mediator of the potential bidirectional relationship between self-esteem and burnout symptoms.

Although self-esteem level is generally considered to be stable[23,38], it can fluctuate on a daily basis due to its reliance on situational and environmental influences[42]. Self-esteem instability refers to the degree of these momentary fluctuations around a baseline level of self-esteem (i.e. higher standard deviation of multiple state self-esteem scores), with unstable self-esteem being associated with greater emotional swings and behavioural reactivity[23]. In a meta-analysis, Okada[63] summarised that although self-esteem level and instability are distinct constructs, they are weakly but negatively correlated and interactively predict various outcomes: Low and stable self-esteem is associated with self-doubt[64] and depression[65], whereas the opposite was found for individuals with high and stable self-esteem[64,66]. In an experimental study[64], individuals with high but unstable self-esteem have been shown to react with self-doubt and higher cardiovascular responses when faced with failure. Their defensive and devaluing responses to criticism[67–69] and lower engagement when faced with challenges (e.g. university exams) may serve as a self-protective (and self-doubt regulating) strategy[64]. Therefore, self-esteem instability might play a moderating role, as individuals with high but unstable self-esteem are more likely to engage in repetitive negative thinking than individuals with high and stable self-esteem.

Researchers have suggested that self-report questionnaires, especially those that retrospectively assess past experiences or general tendencies and traits, may capture a reconstructed rather than an actual experience. Such reports may be influenced by time, (meta)cognitive beliefs and momentary states, leading to potential biases[70,71]. For example, results from two intensive longitudinal studies showed strong discrepancies in worry and rumination between retrospective or global self-reports and real-time assessments of actual experience[72,73]. In addition, there is growing evidence that associations found in between-person analyses do not necessarily reflect mental processes occurring within individuals[74,75]. Between-person variation often arises from different processes than within-person fluctuations over time[74,75]. In particular, cross-sectional studies capture a mixture of trait- and state-level effects, with state-level effects including occasion-specific influences and trait x occasion interactions[76]. Therefore, to better understand how specific modifiable psychological traits (e.g. self-esteem) and self-regulatory strategies (e.g. repetitive negative thinking) affect burnout, an intensive longitudinal design that distinguishes between within-person and between-person effects is needed.

The current ecological momentary assessment study across 4 weeks aimed to (1) examine the relationship between low self-esteem and subsequent daily burnout symptoms, and (2) examine the within-person and between-person mediating role of rumination and pre-sleep worry for the relationship between low self-esteem and burnout. Furthermore, (3) to examine potential bidirectional effects, we tested whether higher levels of burnout would predict an increase in repetitive negative thinking, which would then be associated with lower self-esteem (i.e. the reverse temporal order of Hypotheses 1 and 2). Finally, we (4) explore the moderating role of self-esteem instability on all the relationships stated for Hypotheses 1 and 2.

Based on theory and previous findings, we expected that during the day, self-esteem (i.e. at the morning and noon prompt of each day) would negatively predict subsequent burnout symptoms (i.e. at the noon and evening prompt of each day, respectively), partly mediated by rumination between the respective prompts (at the within-person level; Hypothesis 1). We also hypothesised that the level of self-esteem on the previous day negatively predicted burnout on the following day, partially mediated by pre-sleep worry in the evening (at the within-person level; Hypothesis 2). In addition, we aimed to explore the bidirectional relationships and whether self-esteem instability plays a moderating role for the associations between the level of self-esteem, repetitive negative thinking, and burnout, as well as the proposed mediations (see Hypotheses 1 and 2).

## Methods
### Design and procedure
The present study was approved by the Ethics Committee of Bielefeld University (reference number: 2022-280). Data from this study have already been published in another manuscript with a different topic[77]. Data collection took place during university examination periods between January 2023 and February 2024. Participants were recruited online via social media, email distribution lists from various German universities, and the study participation management portal of the Department of Psychology at Bielefeld University.

Upon registration, participants received detailed information regarding the study procedure, participation conditions, and their rights, including data protection regulations and the option to withdraw at any time. All participants provided informed consent and confirmed their eligibility based on the following inclusion criteria: (a) being between 18 and 30 years old, (b) fluency in German, (c) access to a smartphone with an Android operating system for the study duration (with the alternative option of using a lab-provided phone), and (d) participation in at least one oral or written university examination during the four-week study period.

Each study batch lasted 29 days. On the first day, participants completed a baseline assessment. Ecological momentary assessment commenced on the second day using the movisensXS app (movisens GmbH, Karlsruhe, Germany). Participants received three daily survey prompts over 28 consecutive days: at 8:30 a.m. on weekdays and 9:30 a.m. on weekends for the first survey, followed by prompts at 1:30 p.m. and 8:30 p.m. The response window for each survey was 60 min, after which non-responses were automatically marked as missing. Participants were instructed to disregard survey prompts in situations where responding could pose a safety risk (e.g. while driving). The study design and the investigated hypotheses are illustrated in Fig. 1.

### Participants
A total of 96 individuals participated in the study (78 female, 14 male, 4 diverse or preferred not to disclose their gender), with a mean age of 22.4 years ($SD$ = 2.8, range = 18–30 years). Participants completed 6643 out of 8064 possible surveys, yielding a response rate of 82.4%.

We conducted exploratory analyses testing several potential predictors of missing assessments. These included daytime and weekday of prompt delivery, as well as participants' age, gender and baseline levels of depressive symptoms, anxiety symptoms, somatic symptoms and global self-esteem. None of these variables significantly predicted missingness.

### Measures and instruments
**Demographic information**. Demographic information was assessed within the baseline assessment. Participants were asked with what gender they identify ('What gender do you identify with?'; response options: 'female', 'male', 'diverse', and 'prefer not to respond'). We did not collect data on race or ethnicity.

**Self-esteem**. Momentary self-esteem was assessed in each ecological momentary assessment survey using a single item adapted from the German Single-Item Self-Esteem Scale[78]. Participants rated their agreement with the statement 'I currently have high self-esteem' on a visual analogue scale ranging from 0 (not at all true) to 100 (very true). The subject's mean of self-esteem score on the single item correlated strongly ($r$ = 0.68, $p$ < 0.001) with the baseline score on the Rosenberg Self-Esteem Scale[79,80]. To quantify self-esteem instability, we calculated the within-person standard deviation of momentary self-esteem ratings across all prompts for each participant. Higher values reflect greater instability in self-esteem over time.

**Repetitive negative thinking**. Repetitive negative thinking in the form of pre-sleep worry was measured in the first questionnaire of each day. Participants were asked to indicate on a visual analogue scale (0 = not at all, 100 = very much) to what extent they were worried or upset before sleep ('Were you worried or upset about something before you went to sleep last night?').

Repetitive negative thinking in the form of rumination throughout the day was assessed using a rumination scale specifically developed for ecological momentary assessment studies[81]. The scale consists of five items capturing key characteristics of rumination: perseveration, brooding, negativity, self-criticism, and repetition. Items were rated on a visual analogue scale from 0 (none at all) to 100 (very much). The items were assessed twice daily—at midday and in the evening. At midday, the items assessed rumination experienced since waking, while the evening survey captured rumination occurring since the midday assessment. A mean score was calculated across the five items. The internal consistencies across all measurement points were $\omega_{between}$ = 0.94 and $\omega_{within}$ = 0.88.

**Burnout**. In each ecological momentary assessment survey, participants rated the extent to which they experienced burnout symptoms using a visual analogue scale (0 = not at all, 100 = very much) in response to the question: 'How … do you feel at the moment?'. Burnout, in particular, the key symptom of exhaustion[22], was assessed with the items stressed, exhausted and energetic. To ensure consistency in interpretation, scores for energetic were reversed so that higher values aligned with greater burnout symptom severity. A mean burnout symptom score was computed for each survey. The internal consistencies across all measurement points were $\omega_{between}$ = 0.83 and $\omega_{within}$ = 0.62.

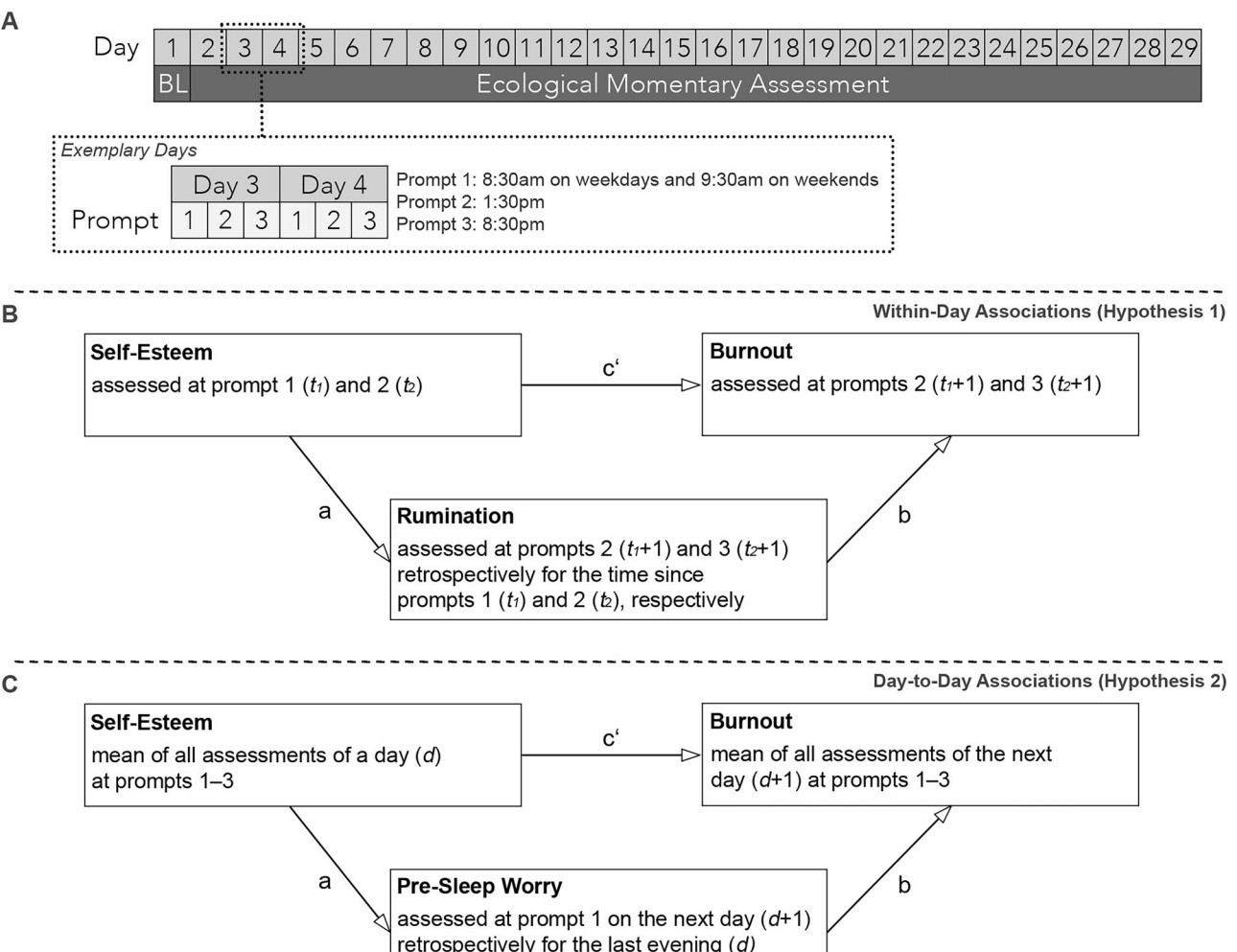

**Fig. 1 | Illustration of the study design and hypotheses.** BL baseline questionnaire. **A** Study design with an exemplary depiction of 2 days which can be generalised to all other days. **B** Conceptual illustration of the within-day mediation analyses for Hypothesis 1. **C** Conceptual illustration of the day-to-day mediation analyses for Hypothesis 2. Please note that the analyses scheme for research question 3 (i.e. the reversed temporal order of self-esteem and burnout) is not depicted here.

## Statistical analyses

Data pre-processing and all statistical analyses were performed using R (Version 4.4.3; R Core Team, 2025). The datasets used and/or analysed during the current study, as well as the analysis code, are available from the corresponding author on reasonable request. The study and analysis plan were not preregistered.

Mediation analysis was performed using the R-package mediation (Version 4.5.0[82]). For a model-based causal mediation analysis, two models need to be specified: (1) a mediator model which includes the mediator variable as the outcome and the independent variable as a predictor to test path a; and (2) an outcome model which includes the dependent variable as the outcome and the independent variable and the mediator variable as predictors to test paths b and c'. To account for the nested structure of our data, we used multilevel models. The models were fitted using the lmer-function from the R-package *lme4* (Version 1.1-35.3[83]) with maximum likelihood estimation. With regard to missing data, the default behaviour of the lmer-function of the *lme4*-package was applied. To disentangle within- and between-person effects, we followed the procedure outlined by Wang and Maxwell[84]. Time-varying predictor variables in each model were decomposed into their within- and between-person components and analysed as separate predictors. The between-person component was obtained by calculating each individual's mean of the respective predictor variable, which was then grand-mean centred. The within-person component was obtained by centreing the predictor variables within each subject. Random

slopes for time-varying predictors were initially specified, but most models failed to converge. Therefore, only random intercepts were included. Gender and age were included as covariates in all models. Data distribution was assumed to be normal, but this was not formally tested.

To test Hypothesis 1, we used current self-esteem at prompt $t$ as a predictor of rumination reported at prompt $t+1$ for the period between $t$ and $t+1$ in the mediator model. In the outcome model, self-esteem at prompt $t$ and rumination reported for the period between $t$ and $t+1$ were included as predictors of burnout symptoms at prompt $t+1$.

To test Hypothesis 2, in the mediator model, we used the mean self-esteem score on day $d$ as a predictor of pre-sleep worry reported retrospectively on the morning of the following day, $d+1$, for the previous night. In the outcome model, the daily mean self-esteem score of day $d$ and pre-sleep worry were included as predictors of the mean burnout symptom score on day $d+1$.

As a further exploratory analysis, we examined the reverse scenario for Hypotheses 1 and 2, where burnout symptoms predict self-esteem and the relationship is mediated by repetitive negative thinking. Corresponding to the analyses of Hypothesis 1, we used current burnout symptoms at prompt $t$ as a predictor of rumination reported at prompt $t+1$ for the period between $t$ and $t+1$ in the mediator model. In the outcome model, burnout symptoms at prompt $t$ and rumination reported for the period between $t$ and $t+1$ were included as predictors of self-esteem at prompt $t+1$. Corresponding to Hypothesis 2, in the mediator model, we used the mean

**Table 1 | Demographics and descriptive statistics**

|  | *N* = 96 | |
|---|---|---|
|  | *M* (SD)/*n* (%) | Min – Max |
| *Demographics* | | |
| Gender | | |
| Female | 78 (81.2) | - |
| Male | 14 (14.6) | - |
| Diverse or no response | 4 (4.2) | - |
| Age | 22.4 (2.8) | 18–30 |
| *Ecological ambulatory assessment* | | |
| Self-esteem | 56.0 (24.2) | 0–100 |
| Self-esteem instability[a] [*SD*] | 11.9 (4.6) | 1.5–23.3 |
| Rumination | 26.9 (21.9) | 0–100 |
| Pre-sleep worry | 32.1 (29.0) | 0–100 |
| Burnout | 46.3 (20.0) | 0–100 |
| Stressed | 36.3 (26.9) | 0–100 |
| Exhausted | 46.3 (28.1) | 0–100 |
| Energetic[b] | 56.4 (23.9) | 0–100 |

[a]The self-esteem instability is based on the computation of the standard deviation of all self-esteem ratings per participant. Higher values indicate a higher instability.
[b]The values were reverse coded to align with the measurements of feeling stressed and exhausted (i.e. higher values indicate feeling less energetic).

burnout score on day *d* as a predictor of pre-sleep worry reported retrospectively on the morning of the following day, *d* + *1*, for the previous night. In the outcome model, the daily mean burnout score of day *d* and pre-sleep worry were included as predictors of the mean self-esteem score on day *d* + *1*.

For the exploratory investigation of the moderating effects of self-esteem instability on the within-person associations between self-esteem, repetitive negative thinking, and burnout, the models described for Hypotheses 1 and 2 were extended by including interaction terms between respective predictors and self-esteem instability. Moderating effects on the mediation analysis were determined by subtracting the respective effect values between groups of different self-esteem instability values (i.e. *M* -1 SD vs. *M* + 1 SD). A bootstrap procedure with 1000 resamples with subsequent confidence interval computation was applied to test the significance of effect differences.

### Reporting summary
Further information on research design is available in the Nature Portfolio Reporting Summary linked to this article.

## Results
### Descriptive statistics
The demographic characteristics and descriptive statistics of the baseline questionnaires and ecological momentary assessment surveys are shown in Table 1.

### Within-day associations of self-esteem, repetitive negative thinking, and burnout
In the mediator model, higher self-esteem than usual was followed by less rumination than usual during the period until the next prompt (i.e. at the within-person level; *b* = −0.23, 95% CI [−0.26; −0−20], *p* < 0.001). Higher self-esteem was also associated with less rumination at the between-person level (*b* = −0.46, 95% CI [−0.60, −0.33], *p* < 0.001). In the outcome model, higher self-esteem than usual was followed by feeling less burnt out than usual at the next prompt (i.e. at the within-person level; *b* = −0.09, 95% CI [−0.12, −0.06], *p* < 0.001). Higher self-esteem was also associated with feeling less burnt out at the between-person level (*b* = −0.24, 95% CI [−0.37, −0.11], *p* < 0.001). Furthermore, more

rumination than usual in between two prompts was followed by feeling more burnt out than usual (i.e. at the within-person level; *b* = 0.30, 95% CI [0.27, 0.33], *p* < 0.001). More rumination was also associated with feeling more burnt out at the between-person level (*b* = 0.29, 95% CI [0.13, 0.44], *p* < 0.001).

The within-person mediation analysis showed a significant indirect effect (*b* = −0.07, 95% CI [−0.08, −0.06], *p* < 0.001), direct effect (*b* = −0.09, 95% CI [−0.12, −0.06], *p* < 0.001), and total effect (*b* = −0.16, 95% CI [−0.19, −0.13], *p* < 0.001) with a proportion of 42% of the total effect being mediated by rumination. The between-person mediation analysis showed a significant indirect effect (*b* = −0.13, 95% CI [−0.22, −0.06], *p* < 0.001), direct effect (*b* = −0.24, 95% CI [−0.37, −0.12], *p* < 0.001), and total effect (*b* = −0.37, 95% CI [−0.49, −0.26], *p* < 0.001) with a proportion of 36% of the total effect being mediated by rumination. The results of the mediation analyses are displayed in Fig. 2.

### Day-to-day associations of self-esteem, repetitive negative thinking, and burnout
In the mediator model, higher self-esteem than usual was followed by less pre-sleep worry than usual (i.e. on the within-person level; *b* = −0.35, 95% CI [−0.43, −0.27], *p* < 0.001) Higher self-esteem was also associated with less pre-sleep worrying at the between-person level (*b* = −0.34, 95% CI [−0.52, −0.17], *p* < 0.001). In the outcome model, higher self-esteem than usual was followed by feeling less burnt out than usual on the next day (i.e. at the within-person level; *b* = −0.04, 95% CI [−0.08, −0.00], *p* = 0.038). Higher self-esteem was also associated with feeling less burnt out at the between-person level (*b* = −0.26, 95% CI [−0.36, −0.16], *p* < 0.001). Furthermore, more pre-sleep worrying than usual was followed by feeling more burnt out than usual (i.e. on the within-person level; *b* = 0.09, 95% CI [0.07, 0.11], *p* < 0.001). More pre-sleep worrying was also associated with feeling more burnt out at the between-person level (*b* = 0.30, 95% CI [0.19, 0.41], *p* < 0.001).

The within-person mediation analysis showed a significant indirect effect (*b* = −0.03, 95% CI [−0.04, −0.02], *p* < 0.001), direct effect (*b* = −0.04, 95% CI [−0.08, −0.00], *p* = 0.042), and total effect (*b* = −0.07, 95% CI [−0.11, −0.03], *p* < 0.001) with a proportion of 44% of the total effect being mediated by rumination. The between-person mediation analysis showed a significant indirect effect (*b* = −0.10, 95%-CI [−0.17, −0.05], *p* < 0.001), direct effect (*b* = −0.26, 95% CI [−0.36, −0.16], *p* < 0.001), and total effect (*b* = −0.36, 95% CI [−0.46, −0.27], *p* < 0.001) with a proportion of 28% of the total effect being mediated by rumination. The results of the mediation analyses are displayed in Fig. 2.

### Exploratory analysis of burnout as a precursor of self-esteem
**Within-day associations.** In the mediator model, higher burnout than usual was followed by more rumination than usual during the period until the next prompt (i.e. at the within-person level; *b* = 0.15, 95% CI [0.12; 0.19], *p* < 0.001). In the outcome model, higher burnout than usual was followed by lower self-esteem than usual at the next prompt (i.e. at the within-person level; *b* = −0.13, 95% CI [−0.16, −0.10], *p* < 0.001). Furthermore, more rumination than usual in between two prompts was followed by lower self-esteem than usual (i.e. at the within-person level; *b* = −0.41, 95% CI [−0.44, −0.38], *p* < 0.001).

The within-person mediation analysis showed a significant indirect effect (*b* = −0.06, 95% CI [−0.08, −0.05], *p* < 0.001), direct effect (*b* = −0.13, 95% CI [−0.16, −0.10], *p* < 0.001), and total effect (*b* = −0.19, 95% CI [−0.23, −0.16], *p* < 0.001) with a proportion of 33% of the total effect being mediated by rumination.

**Day-to-day associations.** In the mediator model, more burnout than usual was followed by more pre-sleep worry than usual (i.e. on the within-person level; *b* = 0.44, 95% CI [0.35, 0.53], *p* < 0.001). In the outcome model, more burnout than usual was followed by lower self-esteem than usual on the next day (i.e. at the within-person level; *b* = −0.12, 95% CI [−0.17, −0.08], *p* < 0.001). Furthermore, more pre-sleep worrying than

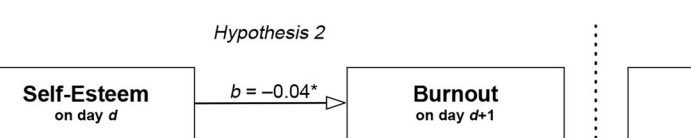

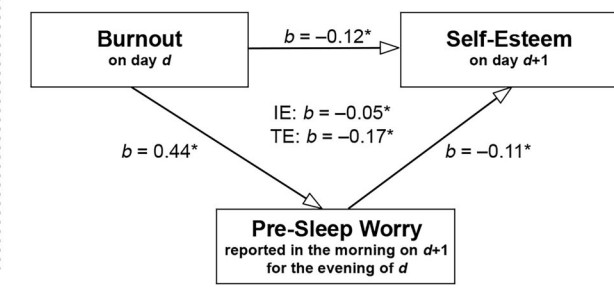

**Fig. 2 | Results of the within-subject mediation analyses.** IE indirect effect, TE total effect, RQ research question; *$p < 0.05$. Ninety-six participants were included in the analyses.

usual was followed by lower self-esteem than usual (i.e. on the within-person level; $b = -0.11$, 95% CI [−0.13, −0.09], $p < 0.001$).

The within-person mediation analysis showed a significant indirect effect ($b = -0.05$, 95% CI [−0.06, −0.03], $p < 0.001$), direct effect ($b = -0.12$, 95% CI [−0.17, −0.08], $p < 0.001$), and total effect ($b = -0.17$, 95% CI [−0.21, −0.13], $p < 0.001$) with a proportion of 28% of the total effect being mediated by rumination.

### Self-esteem instability as a moderator

The within-day associations between self-esteem, repetitive negative thinking, and feeling burnt out were not moderated by the self-esteem instability. However, for the day-to-day within-person associations we found that the association between self-esteem throughout the day and subsequent pre-sleep worry ($b = -0.42$, 95% CI [−0.53, −0.32], $p < 0.001$) was moderated by self-esteem instability ($b = 0.12$, 95% CI [0.02, 0.21], $p = 0.016$) indicating that the association was attenuated for individuals with higher self-esteem instability. Furthermore, the association between pre-sleep worry and next day's burnout ($b = 0.09$, 95% CI [0.07, 0.11], $p < 0.001$) was also moderated by self-esteem instability ($b = 0.03$, 95% CI [0.01, 0.05], $p = 0.011$) indicating that the association was amplified for individuals with higher self-esteem instability. Johnson-Neyman plots of the moderation analyses are displayed in Fig. 3. The mediation analysis for different value ranges of self-esteem instability (i.e. $M$−1 SD vs. $M$ + 1 SD) revealed that neither the indirect effect nor the direct effect significantly differed between individuals with lower and higher self-esteem instability.

### Discussion

The current study examined the relationship between low self-esteem and burnout, to characterise the within- and between-person mediating role of repetitive negative thinking for the relationship between low self-esteem and feeling burnt out, and to explore the moderating role of self-esteem instability on all relationships using ecological momentary assessment. Additionally, we explored whether higher levels of burnout may similarly

predict lower self-esteem levels, also mediated by repetitive negative thinking.

Taken together, the findings of our study suggest that low self-esteem predicts higher burnout symptoms, and that this relationship is partially mediated by repetitive negative thinking (rumination during the day and worrying at bedtime) at the within-person and between-person levels. We also found evidence that higher levels of burnout lead to lower self-esteem levels, partially mediated by repetitive negative thinking.

Regarding the exploratory examination of the moderating role of self-esteem instability, we found within-person moderating effects on the relationship between self-esteem and pre-sleep worry (but not rumination) and on the relationship between pre-sleep worry (but not rumination) and feeling burnt out. However, no such effects were found for the relationship between self-esteem and feeling burnt out at the within-person level and at the between-person level in general.

### Mediation effect of repetitive negative thinking

During the day, lower self-esteem was found to negatively predict burnout symptoms reported in a subsequent survey on the same day, partially mediated by rumination in the meantime. Regarding day-to-day effects, lower self-esteem on the previous day negatively predicted burnout symptoms on the following day, partially mediated by pre-sleep worry in the evening.

At the between-person level, the findings suggest that students with generally lower self-esteem ruminate/worry more frequently over time, which makes them more susceptible to experience stress, exhaustion and lack of energy. At the within-person level, the findings suggest that if a student has lower self-esteem, they also tend to ruminate more, which then contributes to feeling more burnt out, and that if a student has lower self-esteem during the day, they worry more before going to sleep, which then is associated with increased burnout levels the following day.

These results extend the findings of cross-sectional and longitudinal studies showing positive associations between low self-esteem, repetitive

**Fig. 3 | Johnson-Neyman plots for the moderation analyses of self-esteem instability.** Please note that self-esteem instability was group-mean standardised. Therefore, the values represent deviations from the group mean (i.e. the value 1 implies that the self-esteem instability was one standard deviation above the sample's mean). Ninety-six participants were included in the analyses.

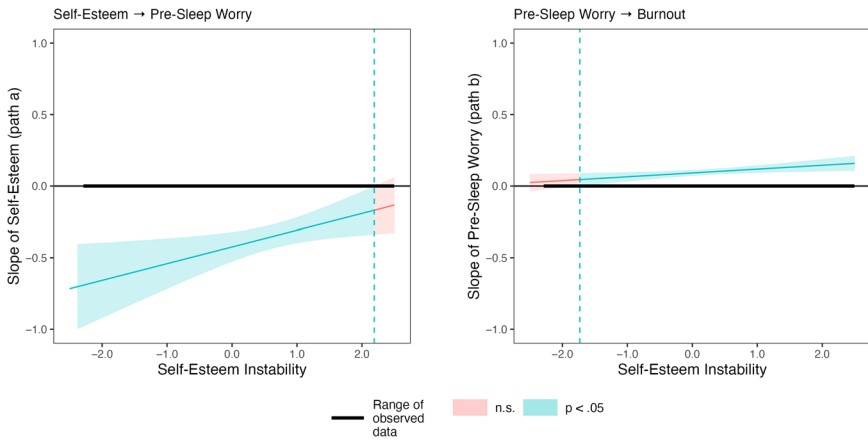

negative thinking and feeling burnt out[24,27,61] by providing evidence that repetitive negative thinking mediated the within-day and day-to-day relationships between low self-esteem and burnout symptoms. Furthermore, the results of our studies are consistent with the Conservation of Resources Theory[31,58], by demonstrating that a lack of recovery experiences through repetitive negative thinking (e.g. pre-sleep worry at bedtime) is associated with ill-being (e.g. burnout) the next day. Given the critical consequences of academic burnout (e.g. increased dropout and impaired long-term wellbeing[3–5]), our findings highlight the potential utility of promoting self-esteem and reducing repetitive negative thinking to mitigate exhaustion and its associated health consequences in university students.

## Burnout as a precursor to low self-esteem

Additionally, we exploratorily investigated whether burnout might also be a precursor to low self-esteem. The findings suggest that higher burnout symptoms during the day (at the morning and noon prompts) also predict lower self-esteem during the day (at the noon and evening prompts), which is partially mediated by rumination between the respective prompts at the within-person level. According to day-to-day associations at the within-person level, our findings suggest that the level of burnout symptoms on the previous day negatively predicts self-esteem on the following day, partially mediated by pre-sleep worry at bedtime.

These findings are in line with evidence from longitudinal tracking methods; for example, ref. 25 found a bidirectional relationship between rumination and self-esteem. Consistent with prior research, repetitive negative thinking not only exacerbates psychological distress but also contributes to its maintenance over time[43,85,86]. Taken together, low self-esteem might contribute to repetitive negative thinking and burnout, and burnout might exacerbate repetitive negative thinking and further erode self-esteem, leading to a vicious cycle.

Future research should address these bidirectional effects on the within-person level. In doing so, it is crucial to acknowledge that many existing experience sampling studies lack a solid theoretical foundation and often provide limited insights into the underlying temporal mechanisms. For example, it is unclear how long individuals must experience low self-esteem to elicit repetitive negative thinking or how sustained such thinking must be to lead to emotional exhaustion. Relatedly, there is a lack of intensive longitudinal studies using experimental designs, that is, for instance, including tailored cognitive micro-interventions to increase or stabilise state self-esteem or to decrease repetitive negative thinking. Such studies are needed to discern the question of causal direction and to better understand the temporal dynamics of the process. In summary, theory-driven and experimental intensive longitudinal research is required in order to clarify these temporal dynamics[87] and examine whether low self-esteem is an initial vulnerability factor, an outcome variable, or whether bidirectional mechanisms are at work within a potential vicious cycle.

## Moderation effect of self-esteem instability

Regarding the moderating role of self-esteem instability, we found mixed results at both the within-person and between-person levels and for rumination and pre-sleep worry. Self-esteem instability moderated the relationship between self-esteem and pre-sleep worry at the within-person level but not at the between-person level, suggesting that a student with highly fluctuating self-esteem benefits less from moments of high self-esteem. Even when they feel worthy, this feeling provides less protection against pre-sleep worrying —perhaps because they remain uncertain about whether the positive state will last. In contrast, a person with more stable self-esteem benefits more from periods of high self-esteem. When they feel worthy, they worry significantly less at bedtime, possibly because their self-esteem is more reliable and consistent. In line with these findings, ref. 88 proposed that individuals with unstable self-esteem are more likely to engage in thinking about the factors contributing to and resulting from momentary self-esteem fluctuations.

In addition, we found evidence for a moderating effect of self-esteem instability on the relationship between pre-sleep worry and burnout at the within-person level but not at the between-person level. When a student's self-esteem was highly unstable, pre-sleep worries were followed by more burnout symptoms than usual during the exam period. This finding is consistent with the results of an experimental study by ref. 64, who found that individuals with unstable self-esteem showed more self-doubt and higher cardiovascular responses, and with ref. 23, who proposed that higher self-esteem instability is associated with greater emotional swings. Students with generally higher self-esteem fluctuations may experience greater emotional and physiological arousal when worrying at bedtime, which could further impair their sleep quality, thereby reducing recovery and promoting exhaustion.

Although self-esteem instability moderated both the a-path (i.e. the association between self-esteem on pre-sleep worry) and the b-path (i.e. the association between pre-sleep worry on next-day burnout symptoms) of the mediation model, these moderating effects were functionally different from a statistical view: Self-esteem instability attenuated the a-path and amplified the b-path. However, as self-esteem instability attenuated the protective effect of higher self-esteem on pre-sleep worry, but amplified the negative impact of pre-sleep worry on burnout, the two moderation effects are similar in that self-esteem instability undermines students' resilience to stress during demanding academic periods in both cases.

In contrast, we did not find evidence for a moderation of the within-day relationships between self-esteem and rumination and between rumination and feeling burnt out by self-esteem instability. Furthermore, no evidence for moderating effects was found for the between-person relationships. However, the absence of evidence of these moderation effects should not be interpreted as evidence of their absence.

## Limitations

In addition to the strengths of the study (e.g. intensive longitudinal design to disentangle within- and between-person mechanisms), it is important to note some of its limitations. First, we only assessed exhaustion, as the starting point of the burnout process[9–11], but not disengagement or reduced self-efficacy, which are also characteristics of burnout[7,8]. Furthermore, we assessed exhaustion using a general momentary affect scale to capture subjective experiences of stress, exhaustion and low energy. Although this differs from the usual domain-specific burnout measures, we consider it quite plausible that students' momentary exhaustion levels are to a meaningful degree determined by stress related to university, given that the data were collected during a high-stakes exam period. However, we acknowledge that using a non-specific exhaustion measure may limit the validity and interpretability of our findings, a limitation that should be addressed in future studies. Second, we used only single items to assess levels of self-esteem and pre-sleep worry, which might reduce the reliability, although ref. 89 provides support for the use of single items in intensive longitudinal designs. Third, in order to examine within-day associations, we assessed rumination throughout the morning/noon and burnout at the same time (i.e. noon, evening), which may have introduced bias. Fourth, it is unclear whether the data were missing completely at random or not, which could have introduced bias. Fifth, using cluster mean centreing in combination with lagged variables could have biased the results[90]. Sixth, our sample consisted mainly of women ($n = 78$; 81.2%), which makes it difficult to generalise the results to all genders. In addition, the results may not be generalised to samples other than university students and to periods other than the exam period. Nevertheless, it is reasonable to assume that the internal vulnerability mechanisms underlying burnout in students and, for example, individuals in helping professions (where burnout is most commonly studied[91]) may be structurally similar (involving an important role of low self-esteem and repetitive negative thinking), despite the different nature of external stressors experienced by these groups (e.g. academic versus interpersonal demands). Future studies need to examine whether similar patterns of associations also hold in different kinds of social, professional, or age groups.

## Conclusion

In summary, our results indicate that although the effect sizes were often much larger at the between-person level, the direction and statistical significance of the associations between individuals are broadly consistent with the associations within individuals. Consequently, prevention programmes aimed at reducing the risk of student burnout and on promoting self-esteem level should focus on reducing past-oriented rumination throughout the day and future-oriented worry at bedtime (e.g. by training mindfulness and strengthening helpful coping mechanisms).

## Data availability

The data used and/or analysed during the current study are available on the Open Science Framework under the files tab (https://doi.org/10.17605/OSF.IO/EXHWD).

## Code availability

The analysis code can be found on the Open Science Framework under the files tab (https://doi.org/10.17605/OSF.IO/EXHWD).

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

## Authors contributions

M.B.: Conceptualisation, methodology, formal analysis, writing—original draft, writing—review and editing; J.H.: Conceptualisation, data curation, formal analysis, investigation, methodology, project administration, writing—original draft, writing—review and editing; E.W.: writing—review and editing; S.L.: Conceptualisation, methodology, project administration, supervision, writing—review and editing.

## Funding

## Competing interests

The authors declare no competing interests.
