## [Transparent Peer Review file · Communications Psychology]

Repetitive Negative Thinking Mediates the Relationship Between Self-Esteem and Burnout in an Ecological Momentary Assessment Study

Corresponding Author: Ms Malin Brueckmann

Version 0:

Decision Letter:

Dear Ms Brueckmann,

Thank you for your patience during the peer-review process. Your manuscript titled "Repetitive Negative Thinking Mediating the Relationship Between Self-Esteem and Burnout in an Ecological Momentary Assessment Study" has now been seen by 2 reviewers, and I include their comments at the end of this message. They find your work of interest but raised some important points. We are interested in the possibility of publishing your study in Communications Psychology, but would like to consider your responses to these concerns and assess a revised manuscript before we make a final decision on publication.

We therefore invite you to revise and resubmit your manuscript, along with a point-by-point response to the reviewers. Please highlight all changes in the manuscript text file.

Editorially, we consider it important that the revised manuscript provide evidence for the validity of the burnout measure as well as more extensive consideration and discussion of its limitations. Furthermore, please provide additional theoretical support for the directional modeling of effects and conduct additional analyses in line with Reviewer 2's suggestions of effects running in the opposite direction or bidirectionality.

I am attaching an Editorial Requests Table that details critical reporting requirements for the revised manuscript. Please attend to each item and ensure your manuscript is fully compliant. If your revised manuscript is not aligned with these requests on major issues, such as those concerning statistics, it may be returned to you for further revisions without re-review.

Please submit the following items:

- Revised manuscript
- Point-by-point response to the referees' comments
- Cover letter (as a separate document)
- <https://www.nature.com/documents/nr-reporting-summary.pdf>>Nature Research Reporting Summary
- Completed Editorial Request Table (attached).

via this link: Link Redacted .

Additional guidance is available in our style and formatting guide Communications Psychology formatting guide.

Best regards,

Troby Lui, on behalf of

Hannah Hao

Troby Lui, PhD
Associate Editor
Communications Psychology

Hannah Hao, PhD
Editorial Board Member
Communications Psychology
orcid.org/0000-0002-3342-9132

REVIEWER EXPERTISE:

Reviewer #1: burnout, EMA
Reviewer #2: burnout

REVIEWER REPORTS:

Reviewer #1 (Remarks to the Author):

In my opinion, a very elegant research design. Great implementation of ecological momentary analysis. Getting real time data at so many different time intervals is incredibly valuable.

In terms of measuring self-esteem and burnout, it would be helpful to more deeply understand how those constructs were assessed. Based on the method section, it seemed that self-esteem was measured with a simple one item question. Is that correct? In the case of burnout, it did seem there were multiple items assessing various elements of the construct, but how many items were there and were those constructs validated?

It seems this data was collected with students. It might be helpful to consider this through that contextual lens. Are there unique aspects about the students lives? With the same findings apply to nurses or social workers or other helping professions (where burnout is most extensively studied).

As a reader, I found myself wanting more information and understanding about the rumination. That seem to be the most interesting component. Are there things people could do to help stop them from ruminating? If one has low self-esteem and begins to spiral with negative thoughts are there ways to mediate those thoughts? Much of the paper discussed self-esteem, which is much more difficult to influence, but in the case of ruminating thoughts, one might be able to cognitively mediate.

Reviewer #2 (Remarks to the Author):

Overall, the paper has numerous strengths, including hard-to-measure ecological momentary assessment data, and statistical analyses that attempt to disentangle causal directions for mediation models. The subject matter is of general interest and importance to the literature. That said, there are a few areas for improvement, which I have enumerated below.

In my view the most important points to address are #1, #5 and #6 below.

1. Core to your overall argumentation in the introduction is that self-esteem is a risk/vulnerability factor, rather than an outcome (e.g., page 4, lines 76-79). This informs not only your hypotheses later, but also the data analysis strategy you ultimately use. However, self-esteem itself is a malleable construct that could just as easily be considered to be an outcome. For instance, meta-analyses suggest it can be increased experimentally (Niveau et al., 2021), and that self-esteem is often an outcome of academic performance rather than the reverse (Baumeister et al., 2005). Moreover, in one of the references you cite (Li et al., 2024), the relationship between self-esteem and rumination is bidirectional. With this in mind, more theoretical development needs to be done in the introduction to support the proposed causal direction of self-esteem to burnout and rumination. That is, why is self-esteem the risk factor, rather than an outcome of burnout and rumination (which seems equally probable). If the direction of the relationship can't be better supported by theory, you may need to re-think your analysis strategy to better account for the possibility of bidirectional relationships with self-esteem (both with rumination and burnout).

Baumeister, R. F., Campbell, J. D., Krueger, J. I., & Vohs, K. D. (2005). Exploding the self-esteem myth. *Scientific American*, 291(1), 84-91. <https://www.jstor.org/stable/10.2307/26060842>

Li, X., He, Y., Ye, K., & Xu, H. (2024). The Bidirectional Relationship between Rumination and Self-Esteem: Evidence from Longitudinal Tracking and Diary Methods. *Studia Psychologica*, 66(3), 193-206. <https://doi.org/10.31577/sp.2024.03.900>

Niveau, N., New, B., & Beaudoin, M. (2021). Self-esteem interventions in adults—a systematic review and meta-analysis. *Journal of Research in Personality*, 94, Article: 104131. <https://doi.org/10.1016/j.jrp.2021.104131>

2. On page 5, lines 94-95 you say: “The self esteem of more than 95% of people is dependent on success and failure”. This statement seems unsupported by the references provided; consider deleting this sentence or providing more precisely the method by which the “more than 95%” statistic is derived.

3. Following your logic for the “Stress as Offense to Self theory” (page 5, lines 92-100), the overall logic of this argument does not entirely make sense to me. You argue that self-esteem threats cause negative emotions. You then argue that the possibility of failure threatens self-esteem during studying, which in turn can lead to negative emotions, which finally leads to burnout. Then you conclude that low trait-level self-esteem predicts the development of burnout. This does not follow logically, because a threat to self-esteem (e.g., an event where a person’s positive self-views are challenged) is not the same thing as the amount of trait self-esteem (i.e., the degree to which people feel positively about themselves). Beyond this, you have not considered and refuted the obvious alternative: Students experiencing burnout are unable to achieve their goals, and thus like and value themselves less (i.e., burnout causes lower self-esteem). Overall, this section was not a strong argument to support self-esteem as a causal factor for burnout.

4. On lines 182-183, you describe that these data have been used in a previous study. Though this study uses pre-sleep worry, the overall hypothesis does seem different enough to merit a different publication. One small query however: In the previous paper, you report two studies with sample sizes $n = 91$ and $n = 129$. This differs from the current study which has $N = 96$. Can you account for this discrepancy?

5. On lines 208-209, you describe the missing data briefly. Can you elaborate somewhere in the paper on how you handled missing data in this paper in subsequent analyses? Moreover, if some manner of imputation process was used, did you evaluate whether other measured variables could predict missingness, and incorporate them in the imputation process (i.e., the missing at random assumption)? Missing data is common in EMA research, but should ideally be handled with some manner of gold standard approach (e.g., full information maximum likelihood, multiple imputation). The default of lme4 is to delete observations with missing data, which may bias the estimated with the amount of missing data present in this study. Multiple imputation is a bit challenging to implement, but I found the Grund et al. (2016) tutorial paper to be very helpful when I had to do it in a recent study of mine, since you’re using R:

Grund, S., Lüdtke, O., & Robitzsch, A. (2016). Multiple imputation of multilevel missing data: an introduction to the R package pan. *Sage Open*, 6(4). <https://doi.org/10.1177/2158244016668220>

6. The operationalization of burnout (lines 233-241) appear only to measure the “exhaustion” facet of the construct. However, the cited model in your paper (Maslach et al., 2001) describes three dimensions “...overwhelming exhaustion, feelings of cynicism and detachment from the job, and a sense of ineffectiveness and lack of accomplishment (pg. 399, Maslach et al., 2001)” Concessions to reduce survey length are inevitable in EMA research, but could you speak to why you focused on this facet of burnout specifically in this study? I suspect it is because the broader study was about sleep, but there may be other theoretical considerations that went into your decision. You acknowledge the other two dimensions of burnout in the limitations section, but don’t elaborate or unpack it.

Beyond this, I’m not entirely sure that the face validity of these 3 items (stressed, exhausted, energetic) necessarily captures the exhaustion facet of burnout, which is normally measured by items like these on the MBI:

I feel emotionally exhausted because of my work

I fell worn out at the end of a working day

I feel tired as soon as I get up in the morning and see a new working day stretched out in front of me.

Working with people the whole day is stressful for me

That is, burnout exhaustion is specifically exhaustion caused by work (similar items have been constructed for academic burnout; see Salmela-Aro et al., 2009). In contrast, the way you ask the items is just “how ... do you feel at the moment.” Your measure seems more like a general affect measure, mixing fatigue and stress together. That is, how do you know that you are really measuring “burnout” with this measure, rather than simply a fatigue subdimension of negative affect?

Salmela-Aro, K., Kiuru, N., Leskinen, E., & Nurmi, J. E. (2009). School burnout inventory (SBI) reliability and validity. *European journal of psychological assessment*, 25(1), 48-57.

7. This is minor, but could you describe how self-esteem instability was measured in the method? It is described in a footnote in Table 1, but probably would fit better under “self-esteem” in the materials, or otherwise in the data analysis strategy so that it is clear when reading the paper in order from start to finish.

8. The analysis strategy in general seems well thought out, though it is somewhat unusual. I have seen models where (a) variance is partitioned into between vs. within components with person mean centering and (b) lagged models where scores at t predict $t+1$ separately, but I have not seen them combined together in a single model as you have here before. Could you walk me through the rationale for doing the model in this way, and/or direct me to any prior research that analyzed data via this method? Why do both (a) and (b) instead of just one or the other? You cite Wang & Maxwell (2015), but I think they only talk about (a). I suspect this would also involve describing the rationale for your study method/design in Figure 1, which is somewhat more complex than simply measuring all variables at all prompts.

9. When reporting slopes for various coefficients, consider also reporting some measure of the slope's variability. Ideally, a 95% confidence interval, but a standard error would also be suitable for this purpose. If you were tight on words, 95% CIs could replace the in-text p-values, especially since most of the p-values are $< .001$.

10. This is optional, but consider figures in the paper (or online supplement) visualizing the interaction effects (e.g., simple slopes plots), as it might be helpful for readers to grasp the overall pattern better.

11. If I'm following the results on the moderation in lines 318-331 properly, it is a somewhat complicated pattern inasmuch as self-esteem instability attenuates the a-path, but amplifies the b-path. Since the interaction term coefficient is larger for the amplification than for attenuation (i.e., coefficients of 0.09 vs. 0.03), the net effect on the indirect effect is amplification (i.e., the indirect effect is larger with higher self-esteem instability). Am I following that right? If so, this could be made a little clearer in the discussion, as you don't talk about the moderated indirect effect at all there (only the a-path and b-path separately).

12. In the limitations section, you describe that future research should address bidirectional relationships. Why didn't you do that in your study? Is it a limitation of the somewhat unusual measurement schedule for variables?

I hope that these comments are helpful in the revision of your manuscript.

Sincerely,

Sean P. Mackinnon
Dalhousie University

Communications Psychology is committed to improving transparency in authorship. As part of our efforts in this direction, we are now requesting that all authors identified as ‘corresponding author’ create and link their Open Researcher and Contributor Identifier (ORCID) with their account on the Manuscript Tracking System prior to acceptance. ORCID helps the scientific community achieve unambiguous attribution of all scholarly contributions. You can create and link your ORCID from the home page of the Manuscript Tracking System by clicking on ‘Modify my Springer Nature account’ and following the instructions in the link below. Please also inform all co-authors that they can add their ORCIDs to their accounts and that they must do so prior to acceptance.

Version 1:

Decision Letter:

Dear Ms Brueckmann,

Your manuscript titled "Repetitive Negative Thinking Mediating the Relationship Between Self-Esteem and Burnout in an Ecological Momentary Assessment Study" has now been seen by our reviewers, whose comments appear below. In light of their advice I am delighted to say that we are happy, in principle, to publish a suitably revised version in Communications Psychology.

We therefore invite you to revise your paper one last time to address the remaining concerns of our reviewers and a list of editorial requests. At the same time we ask that you edit your manuscript to comply with our format requirements and to maximise the accessibility and therefore the impact of your work.

EDITORIAL REQUESTS:

SUBMISSION INFORMATION:

OPEN ACCESS:

*** TRANSPARENT PEER REVIEW:** Communications Psychology uses a transparent peer review system. On author request, confidential information and data can be removed from the published reviewer reports and rebuttal letters prior to publication. If you are concerned about the release of confidential data, please let us know specifically what information you would like to have removed. Please note that we cannot incorporate redactions for any other reasons.

*** CODE AVAILABILITY:** All Communications Psychology manuscripts must include a section titled "Code Availability" at the end of the methods section. We require that the custom analysis code supporting your conclusions is made available in a publicly accessible repository at this stage; please choose a repository that generates a digital object identifier (DOI) for the code; the link to the repository and the DOI must be included in the Code Availability statement. Publication as Supplementary Information will not suffice.

*** DATA AVAILABILITY:**

Link Redacted

Best regards,

Troy Lui, on behalf of

Hannah Hao

Troy Lui, PhD
Associate Editor
Communications Psychology

Hannah Hao, PhD
Editorial Board Member
Communications Psychology
orcid.org/0000-0002-3342-9132

REVIEWERS' COMMENTS:

Reviewer #1 (Remarks to the Author):

My concerns with the initial manuscript were related to how self-esteem and burnout were being measured. Both have been addressed well in this revision.

In addition, I wanted more clarification about how the authors were operationally defining rumination and what recommendations there might be. This has also been adequately addressed in the latest revision.

Reviewer #2 (Remarks to the Author):

Overall, the authors did an excellent job addressing my previous comments and revising the manuscript. They also went above and beyond in some of the analytic components by exploring the reverse direction of lagged relationships and adding JN plots for the interaction. I'm also glad you found the coding error towards the end with the interaction prior to publication, and the results are more sensible now that it is fixed. I have only a few remaining comments to share for you to optionally consider and now consider this paper suitable for publication.

5. I definitely understand not wanting to go the route of multiple imputation. This is already a complex study, and it is quite challenging to implement and might result in uncertain benefits in an analysis like this. That said, I have a couple of small follow-ups.

Though it is a subtle distinction, lme4 discards incomplete observations, not incomplete participants. The term "listwise deletion" usually implies keeping only those participants with complete data on all observations, which isn't quite accurate (and in fact, would be inferior to what lme4 actually does!). See this stack overflow post, and Ben Bolker's response specifically as he played a pivotal role in developing lme4 and knows way more than me:

<https://stackoverflow.com/questions/78636723/mixed-effects-models-does-lmer-function-really-do-listwise-deletion>

So, all that to say maybe don't say "listwise deletion" as it is somewhat misleading, and describe it more like in this post.

Secondly, your missing data analysis on lines 258-262 is a good addition, though it is inconclusive. This means you don't know if the data are (a) missing completely at random, which means your results are unbiased or (b) missing not at random, based on some unmeasured predictors, which could bias the results. Adding a sentence somewhere in the paper (either around here, or in the limitations section) discussing this uncertainty briefly would be welcome, but is not strictly necessary.

8. You did provide a number of citations to the method of past empirical studies where the combination of group mean centering and lagged variables are used, but I was actually hoping for a statistical reference. That is, I wasn't looking for evidence of what other people have done, but rather, statistical evidence (e.g., from simulation studies) that this approach is desirable. Reviewing all of the studies you cite briefly, it doesn't look like any of them have a statistical reference citing this approach either, so it is possible such a citation does not exist.

I did a little bit of research on the subject, I'd like to just leave you with this citation, suggesting that level 1 autoregressive parameters are biased downwards (i.e., too small) when cluster mean centering is used when compared to uncentered predictors, which is the closest analogue I could find to what you did (cluster mean centering with lagged variables, but not autoregressive ones). To be clear: I'm not asking you to re-analyze any data, but rather just sharing this with you in the

interest of possibly re-thinking the use of cluster mean centering and lagged variables simultaneously in the future in the interest of collegial information sharing, as I think the properties of models with both procedures combined may be unknown at present:

Hamaker, E. L., & Grasman, R. P. (2015). To center or not to center? Investigating inertia with a multilevel autoregressive model. *Frontiers in psychology*, 5, 1492. <https://doi.org/10.3389/fpsyg.2014.01492>

Finally, a two typos I noticed in the revisions you might consider fixing prior to publication:

Line 55: "...Higher levels of burn out" should be "burnout"

Line 122, the direct quotation is missing the page number which I think is normally included

Sean P. Mackinnon
Dalhousie University

Point-by-point response

Dear Reviewer 1, dear Reviewer 2,

we would like to thank you and both reviewers for the encouragement and the valuable and very helpful ideas for improvement. We have carefully addressed all comments and suggestions listed in the reviews, which has clearly strengthened our manuscript. To improve comprehensibility, we provide additional theoretical support for the directional modelling of effects. Furthermore, we conducted additional analyses to determine whether the effects also run in the opposite direction. We have also made substantial changes that have led to a deeper and more critical discussion of our results. For example, we now discuss limitations of our burnout measure in more detail.

We address all individual comments in more detail below.

Please also note that we now use the maximum likelihood estimation and no longer the restricted maximum likelihood estimation when calculating the multilevel models, as we focus on reporting fixed effects. This has led to minimal changes in the coefficients, but not to different results in terms of significance or conclusions.

Different colors were used to indicate reviewer comments (**black**), author responses (**red**), and changes made within the manuscript (**green**). All page numbers refer to the Microsoft Word setting “simple markup”.

We think that our manuscript has benefited substantially from the revision according to the points mentioned in the reviews, and we hope to convince you that this version of the manuscript is now eligible for publication in *Communications Psychology*.

Yours sincerely,

the authors

Reviewer 1

In my opinion, a very elegant research design. Great implementation of ecological momentary analysis. Getting real time data at so many different time intervals is incredibly valuable.

We thank the reviewer for the valuable feedback and the positive evaluation of our manuscript!

In terms of measuring self-esteem and burnout, it would be helpful to more deeply understand how those constructs were assessed. Based on the method section, it seemed that self-esteem was measured with a simple one item question. Is that correct? In the case of burnout, it did seem there were multiple items assessing various elements of the construct, but how many items were there and were those constructs validated?

Thank you for asking such detailed questions.

Momentary self-esteem was indeed assessed using a single item adapted from the German Single-Item Self-Esteem Scale (G-SISE). The validity of this item was demonstrated by Barilovskaia & Margraf (2020), who found evidence of convergence between the G-SISE (“I currently have high self-esteem”) and the German Rosenberg Self-Esteem Scale (Rosenberg, 1965), which is often used to measure self-esteem level. Furthermore, they established test-retest reliability over a nine-month period. In summary, they concluded that the G-SISE is a valid, reliable and economical instrument for measuring global self-esteem. Consistent with this, our study’s findings showed that the subject’s mean of self-esteem score on the single item correlated strongly with their baseline score on the Rosenberg Self-Esteem Scale ($r = .68, p < .001$). We report this correlation in the method section (p. 12, lines 264 to 272).

As Reviewer 2 has also questioned the construct validity of our burnout measure, please refer to point 6 for further discussion.

It seems this data was collected with students. It might be helpful to consider this through that contextual lens. Are there unique aspects about the students lives? With the same findings apply to nurses or social workers or other helping professions (where burnout is most extensively studied).

Thank you for pointing this out. Unlike in traditional contexts of burnout, such as healthcare or social work (O'Connor et al., 2018), where interpersonal demands dominate, student burnout appears to be driven by high academic demands, especially during exam periods. Although the nature of external stressors differs between students and individuals in helping professions, the underlying vulnerability mechanisms may be structurally similar. In both contexts, individuals are often highly invested in their roles and derive a significant portion of their self-esteem from perceived competence (Dahlin et al., 2007; Hallsten et al., 2005; Huang et al., 2025). Therefore, the internal psychological mechanisms, such as lower and more unstable self-esteem, as well as repetitive negative thinking, may create comparable pathways to burnout. We have addressed your suggestion in the limitations section (p. 24, lines 585-593):

In addition, the results may not be generalised to samples other than university students and to periods other than the exam period. Nevertheless, it is reasonable to assume that the internal vulnerability mechanisms underlying burnout in students and, for example, individuals in helping professions (where burnout is most commonly studied⁹¹) may be structurally similar (involving an important role of low self-esteem and repetitive negative thinking), despite the different nature of external stressors experienced by these groups (e.g. academic versus interpersonal demands). Future studies need to examine whether similar patterns of associations also hold in different kinds of social, professional, or age groups.

Dahlin, M., Joneborg, N., & Runeson, B. (2007). Performance-based self-esteem and burnout in a cross-sectional study of medical students. *Medical teacher*, 29(1), 43–48. <https://doi.org/10.1080/01421590601175309>

Hallsten, L., Josephson, M., & Torgén, M. (2005). *Performance-based self-esteem: A driving force in burnout processes and its assessment*.

Huang, H., Zhang, X., Tu, L., Peng, W., Wang, D., Chong, H., ... & Chen, H. (2025). Inclusive leadership, self-efficacy, organization-based self-esteem, and intensive care nurses' job performance: A cross-sectional study using structural equation modeling. *Intensive and Critical Care Nursing*, 87, 103880. <https://doi.org/10.1016/j.iccn.2024.103880>

O'Connor, K., Muller Neff, D., & Pitman, S. (2018). Burnout in mental health professionals: A systematic review and meta-analysis of prevalence and

determinants. *European psychiatry: the journal of the Association of European Psychiatrists*, 53, 74–99. <https://doi.org/10.1016/j.eurpsy.2018.06.003>

As a reader, I found myself wanting more information and understanding about the rumination. That seem to be the most interesting component. Are there things people could do to help stop them from ruminating? If one has low self-esteem and begins to spiral with negative thoughts are there ways to mediate those thoughts? Much of the paper discussed self-esteem, which is much more difficult to influence, but in the case of ruminating thoughts, one might be able to cognitively mediate.

Thank you for this interesting question. Research in cognitive and clinical psychology has shown that interventions such as metacognitive therapy, mindfulness-based cognitive therapy, and cognitive behavioral techniques can effectively reduce ruminative thinking (Hilt & Pollak, 2012; Perestelo-Perez et al., 2017; Querstret & Cropley, 2012). These approaches help individuals learn to recognise when they are entering a ruminative state, to decenter from their thoughts, and to shift their attention towards more adaptive and constructive cognitive or behavioural strategies. For example, interventions that encourage individuals to think more concretely and specifically (Watkins et al., 2009; Watkins et al., 2012) or promote cognitive restructuring towards more positive and constructive interpretations have been shown to reduce rumination and worry. While we fully agree that reducing repetitive negative thinking is a highly relevant area for practical application and further research, we have chosen not to discuss intervention programmes in this manuscript since none were empirically tested in our study. Therefore, we only included the following practical implications in our conclusion (p. 25, lines 598-605):

Consequently, prevention programmes aimed at reducing the risk of student burnout and on promoting self-esteem level should focus on reducing past-oriented rumination throughout the day and future-oriented worry at bedtime (e.g., by training mindfulness and strengthening helpful coping mechanisms). Furthermore, not only students with lower-than-average self-esteem could benefit from prevention programmes, but all students experiencing higher variability in self-esteem (regardless of trait self-esteem levels) could benefit from psychoeducation that normalises daily fluctuations in self-esteem to reduce triggers for repetitive negative thinking.

Hilt, L.M., Pollak, S.D. Getting Out of Rumination: Comparison of Three Brief Interventions in a Sample of Youth. *J Abnorm Child Psychol* **40**, 1157–1165 (2012). <https://doi.org/10.1007/s10802-012-9638-3>

Perestelo-Perez, L., Barraca, J., Penate, W., Rivero-Santana, A., & Alvarez-Perez, Y. (2017). Mindfulness-based interventions for the treatment of depressive rumination: Systematic review and meta-analysis. *International Journal of Clinical and Health Psychology*, *17*(3), 282-295. <https://doi.org/10.1016/j.ijchp.2017.07.004>

Querstret, D., & Cropley, M. (2013). Assessing treatments used to reduce rumination and/or worry: A systematic review. *Clinical psychology review*, *33*(8), 996-1009. <https://doi.org/10.1016/j.cpr.2013.08.004>

Watkins, E. R., Baeyens, C. B., & Read, R. (2009). Concreteness training reduces dysphoria: proof-of-principle for repeated cognitive bias modification in depression. *Journal of abnormal psychology*, *118*(1), 55. <https://doi.org/10.1037/a0013642>

Watkins, E. R., Taylor, R. S., Byng, R., Baeyens, C., Read, R., Pearson, K., & Watson, L. (2012). Guided self-help concreteness training as an intervention for major depression in primary care: a Phase II randomized controlled trial. *Psychological medicine*, *42*(7), 1359-1371. <https://doi.org/10.1017/S0033291711002480>

Reviewer 2

Overall, the paper has numerous strengths, including hard-to-measure ecological momentary assessment data, and statistical analyses that attempt to disentangle causal directions for mediation models. The subject matter is of general interest and importance to the literature. That said, there are a few areas for improvement, which I have enumerated below. In my view the most important points to address are #1, #5 and #6 below.

We thank the reviewer for their feedback and helpful suggestions to improve the robustness of our manuscript.

1. Core to your overall argumentation in the introduction is that self-esteem is a risk/vulnerability factor, rather than an outcome (e.g., page 4, lines 76-79). This informs not only your hypotheses later, but also the data analysis strategy you ultimately use. However, self-esteem itself is a malleable construct that could just as easily be considered to be an outcome. For instance, meta-analyses suggest it can be increased experimentally (Niveau et al., 2021), and that self-esteem is often an outcome of academic performance rather than the reverse (Baumeister et al., 2005). Moreover, in one of the references you cite (Li et al., 2024), the relationship between self-esteem and rumination is bidirectional. With this in mind, more theoretical development needs to be done in the introduction to support the proposed causal direction of self-esteem to burnout and rumination. That is, why is self-esteem the risk factor, rather than an outcome of burnout and rumination (which seems equally probable). If the direction of the relationship can't be better supported by theory, you may need to re-think your analysis strategy to better account for the possibility of bidirectional relationships with self-esteem (both with rumination and burnout).

Baumeister, R. F., Campbell, J. D., Krueger, J. I., & Vohs, K. D. (2005). Exploding the self-esteem myth. *Scientific American*, 291(1), 84-91. <https://www.jstor.org/stable/10.2307/26060842>

Li, X., He, Y., Ye, K., & Xu, H. (2024). The Bidirectional Relationship between Rumination and Self-Esteem: Evidence from Longitudinal Tracking and Diary Methods. *Studia Psychologica*, 66(3), 193-206. <https://doi.org/10.31577/sp.2024.03.900>

Niveau, N., New, B., & Beaudoin, M. (2021). Self-esteem interventions in adults—a systematic review and meta-analysis. *Journal of Research in Personality*, 94, Article: 104131. <https://doi.org/10.1016/j.jrp.2021.104131>

We welcomed this comment and have accordingly expanded the theoretical background (see, pp. 5-6, lines 99-132). In addition to providing further theoretical support for the proposed causal relationship, we now also consider, analyse, and briefly discuss the reverse scenario, which as we believe has substantially strengthened the manuscript (see, pp. 21-22).

According to the Conservation of Resources Theory³¹, which states that stress arises from a perceived threat to resources, self-esteem is considered a key personal resource that buffers stress³². High self-esteem serves as a protective factor, enabling individuals to cope more effectively with high demands, such as those encountered during exam periods. Consistent with this, higher self-esteem has been shown to result in a lower physiological response to challenging or achievement-related situations^{33,34}. In contrast, individuals with lower self-esteem exhibit stronger cortisol responses to stress and tend to perform worse under pressure³⁵. The results of an experimental neuroimaging study further suggest, that individuals with lower self-esteem “might have to put more effort in cognitive performance control, stress and, emotion regulation and, furthermore, think about themselves and relive previous stressful experiences.”³⁴. These increased stress response and regulatory demand may deplete cognitive and emotional resources more quickly, thereby contributing to (emotional) exhaustion.

Thinking further, (emotional) exhaustion, followed by disengagement and reduced self-efficacy⁹ may impair students’ ability to achieve their goals (e.g., due to lower performance³⁶). The findings of an experimental study show that academic failure drains students’ state self-esteem³⁷, leading to a vicious cycle of low (state) self-esteem and burnout symptoms.

Traditionally, self-esteem has been conceptualized as a relatively stable trait^{23,38} and regarded as a key vulnerability factor for psychological distress (e.g., burnout^{39,40}). This perspective is supported by developmental considerations: While self-esteem develops from early childhood onwards, burnout is a later-emerging, context-specific phenomenon that arises in response to chronic role-related stress^{8,22}, especially when important resources are lacking. Consistent with this, empirical research shows that

burnout symptoms increase during lower secondary education, and are closely associated with structured performance environments such as schools, universities, and workplaces^{12,41}. Based on this reasoning, we conceptualise self-esteem as a precursor to burnout development.

At the same time, self-esteem also exhibits meaningful short-term variability⁴². Using a state-based measure of self-esteem in our study allows us to examine its predictive role and to test the reverse direction, i.e., whether daily experiences of stress and exhaustion predict subsequent decreases in self-esteem. Overall, while our study follows the theoretical tradition of modelling self-esteem as a predictor of burnout symptoms, it also acknowledges the potential for bidirectional effects by testing for the possibility that burnout-related states may in turn erode students' self-esteem.

An additional section has been added at the end of the consideration of repetitive negative thinking as a mediator (p. 8, lines 164-173):

Similarly, experiencing burnout symptoms may lead to repetitive negative thinking, which can result in lower (state) self-esteem. In an academic context, students experiencing burnout symptoms, particularly in terms of (emotional) exhaustion and reduced self-efficacy, may ruminate more frequently about past failures or worry about future academic demands. Such repetitive negative thinking can reinforce negative self-perceptions and undermine momentary self-esteem. Longitudinal studies support this, showing bidirectional associations of repetitive negative thinking with both self-esteem²⁵ and burnout symptoms (e.g., exhaustion⁶²). In summary, both theoretical considerations and empirical evidence suggest that repetitive negative thinking may be the mediator of the potential bidirectional relationship between self-esteem and burnout symptoms.

Further changes regarding the discussion (pp. 20-21, lines 476-502):

Burnout as A Precursor to Low Self-Esteem

Additionally, we exploratorily investigated whether burnout might also be a precursor to low self-esteem. The findings suggest that higher burnout symptoms during the day (at the morning and noon prompts) also predict lower self-esteem during the day (at the noon and evening prompts), which is partially mediated by rumination between the respective prompts at the within-person level. According to day-to-day associations at the within-person level, our findings suggest that the level of burnout symptoms on the

previous day negatively predicts self-esteem on the following day, partially mediated by pre-sleep worry at bedtime.

These findings are in line with evidence from longitudinal tracking methods; for example, Li et al.²⁵ found a bidirectional relationship between rumination and self-esteem. Consistent with prior research, repetitive negative thinking not only exacerbates psychological distress but also contributes to its maintenance over time^{86–88}. Taken together, low self-esteem might contribute to repetitive negative thinking and burnout, and burnout might exacerbate repetitive negative thinking and further erode self-esteem, leading into a vicious cycle.

Future research should address these bidirectional effects on the within-person level. In doing so, it is crucial to acknowledge that many existing experience sampling studies lack a solid theoretical foundation and often provide limited insights into the underlying temporal mechanisms. For example, it is unclear how long individuals must experience low self-esteem to elicit repetitive negative thinking, or how sustained such thinking must be to lead to emotional exhaustion. Relatedly, there is a lack of intensive longitudinal studies using experimental designs, that is for instance including tailored cognitive micro-interventions to increase or stabilise state self-esteem or to decrease repetitive negative thinking. Such studies are needed to discern the question of causal direction and to better understand the temporal dynamics of the process. In summary, theory-driven and experimental intensive longitudinal research is required in order to clarify these temporal dynamics⁸⁹ and examine whether low self-esteem is an initial vulnerability factor, an outcome variable, or whether bidirectional mechanisms are at work within a potential vicious cycle.

2. On page 5, lines 94-95 you say: “The self esteem of more than 95% of people is dependent on success and failure”. This statement seems unsupported by the references provided; consider deleting this sentence or providing more precisely the method by which the “more than 95%” statistic is derived.

Thank you for pointing this out, we agree and have therefore removed the statement from the article.

3. Following your logic for the “Stress as Offense to Self theory” (page 5, lines 92-100), the overall logic of this argument does not entirely make sense to me. You argue that self-esteem threats cause negative emotions. You then argue that the possibility of failure threatens self-esteem during studying, which in turn can lead to negative emotions, which finally leads to burnout. Then you conclude that low trait-level self-esteem predicts the development of burnout. This does not follow logically, because a threat to self-esteem (e.g., an event where a person’s positive self-views are challenged) is not the same thing as the amount of trait self-esteem (i.e., the degree to which people feel positively about themselves). Beyond this, you have not considered and refuted the obvious alternative: Students experiencing burnout are unable to achieve their goals, and thus like and value themselves less (i.e., burnout causes lower self-esteem). Overall, this section was not a strong argument to support self-esteem as a causal factor for burnout.

Thank you for examining the logic of the argument so thoroughly and for identifying its flaws. We agree with the reviewer and have therefore rewritten the paragraph (see, pp. 5-6, lines. 99-111). We now use the Conservation of Resources Theory (Hobfoll, 1989) as a theoretical framework, supporting the reasoning with the findings of an experimental neuroimaging study (Kogler et al., 2017). As this critique complements your critique in point 1, you can find the changes there.

Hobfoll S. E. (1989). Conservation of resources. A new attempt at conceptualizing stress. *The American psychologist*, 44(3), 513–524. <https://doi.org/10.1037//0003-066x.44.3.513>

Kogler, L., Seidel, EM., Metzler, H. *et al.* Impact of self-esteem and sex on stress reactions. *Sci Rep* 7, 17210 (2017). <https://doi.org/10.1038/s41598-017-17485-w>

4. On lines 182-183, you describe that these data have been used in a previous study. Though this study uses pre-sleep worry, the overall hypothesis does seem different enough to merit a different publication. One small query however: In the previous paper, you report two studies with sample sizes $n = 91$ and $n = 129$. This differs from the current study which has $N = 96$. Can you account for this discrepancy?

Thank you for this question. A total of 96 people took part in the study presented in this manuscript which corresponds to Study 2 in the previous paper. In the previous paper, five of these people were not included in the analyses as they did not provide complete data for at least one night (i.e., actigraphy data for one night and the

corresponding morning questionnaire). This was mainly due to missing actigraphy data caused by technical defects or loss of the device.

The study with a sample size of $N = 129$, which was reported in a previous paper, is based on different data collection. In this study, neither burnout nor rumination were assessed.

5. On lines 208-209, you describe the missing data briefly. Can you elaborate somewhere in the paper on how you handled missing data in this paper in subsequent analyses? Moreover, if some manner of imputation process was used, did you evaluate whether other measured variables could predict missingness, and incorporate them in the imputation process (i.e., the missing at random assumption)? Missing data is common in EMA research, but should ideally be handled with some manner of gold standard approach (e.g., full information maximum likelihood, multiple imputation). The default of *lme4* is to delete observations with missing data, which may bias the estimated with the amount of missing data present in this study. Multiple imputation is a bit challenging to implement, but I found the Grund et al. (2016) tutorial paper to be very helpful when I had to do it in a recent study of mine, since you're using R:

Grund, S., Lüdtke, O., & Robitzsch, A. (2016). Multiple imputation of multilevel missing data: an introduction to the R package *pan*. *Sage Open*, 6(4). <https://doi.org/10.1177/2158244016668220>

We thank the reviewer for this important comment on the treatment of missing data. As correctly noted, we used the default handling of missing data in the *lme4* package, which entails listwise deletion of observations with missing values. We have added a clarifying statement to the Statistical Analysis section (pp. 13, lines 309-310):

With regard to missing data, listwise deletion was applied for each model, following the default behaviour of the *lmer*-function of the *lme4*-package.

In our dataset, missingness did not occur at the level of individual items but at the level of entire assessment points (i.e., full assessments were missing, not individual variables within an assessment). To address the reviewer's concern, we have now examined whether missingness in assessments could be predicted by other observed variables. Specifically, we examined time-of-day, day-of-week, participant age, gender, and baseline levels of depressive symptoms, anxiety symptoms, somatic

symptoms, and global self-esteem. None of these variables were significant predictors of missingness. We have added a corresponding statement in the Participants section of the manuscript (pp. 11, lines 258-262):

We conducted exploratory analyses testing several potential predictors of missing assessments. These included daytime and weekday of prompt delivery, as well as participants' age, gender, and baseline levels of depressive symptoms, anxiety symptoms, somatic symptoms, and global self-esteem. None of these variables significantly predicted missingness.

As noted by Fritz et al. (2024; <https://doi.org/10.1177/25152459241267912>), distinguishing whether missingness in ecological momentary assessment (EMA) data is completely at random or not at random remains challenging and currently no suitable approaches exist. Given the context of our study – an examination period – it is plausible that many missed prompts occurred due to participants being particularly stressed or highly focused on studying.

We acknowledge the merits of multiple imputation and full information maximum likelihood approaches. However, as also discussed in the literature (Fritz et al., 2024; <https://doi.org/10.1177/25152459241267912>), imputation in the context of EMA or intensive longitudinal data is controversial and seldomly applied. EMA data are characterized by high within-person variability and strong contextual influences. Imputing such momentary experiences may be inappropriate or even illogical. For these reasons, we opted not to use statistical approaches for handling missing data.

6. The operationalization of burnout (lines 233-241) appear only to measure the “exhaustion” facet of the construct. However, the cited model in your paper (Maslach et al., 2001) describes three dimensions “...overwhelming exhaustion, feelings of cynicism and detachment from the job, and a sense of ineffectiveness and lack of accomplishment (pg. 399, Maslach et al., 2001)” Concessions to reduce survey length are inevitable in EMA research, but could you speak to why you focused on this facet of burnout specifically in this study? I suspect it is because the broader study was about sleep, but there may be other theoretical considerations that went into your decision. You acknowledge the other two dimensions of burnout in the limitations section, but don't elaborate or unpack it.

Beyond this, I'm not entirely sure that the face validity of these 3 items (stressed, exhausted,

energetic) necessarily captures the exhaustion facet of burnout, which is normally measured by items like these on the MBI:

I feel emotionally exhausted because of my work

I feel worn out at the end of a working day

I feel tired as soon as I get up in the morning and see a new working day stretched out in front of me.

Working with people the whole day is stressful for me

That is, burnout exhaustion is specifically exhaustion caused by work (similar items have been constructed for academic burnout; see Salmela-Aro et al., 2009). In contrast, the way you ask the items is just “how ... do you feel at the moment.” Your measure seems more like a general affect measure, mixing fatigue and stress together. That is, how do you know that you are really measuring “burnout” with this measure, rather than simply a fatigue subdimension of negative affect?

Salmela-Aro, K., Kiuru, N., Leskinen, E., & Nurmi, J. E. (2009). School burnout inventory (SBI) reliability and validity. *European journal of psychological assessment*, 25(1), 48-57.

Thank you for pointing this out. The broader study was indeed about sleep, and examining burnout was not originally planned. However, many studies define burnout in terms of exhaustion, as it is widely accepted that exhaustion is the primary symptom of burnout (Parker & Salmela-Aro, 2011; Roskam & Mikolajczak, 2021; Taris et al., 2005). We have added two sentences to the introduction to explain the suggested developmental process of burnout, thus clarifying that we consider exhaustion to be its key symptom. Furthermore, Salmela-Aro et al. (2009) propose that the dimensions of burnout are closely linked but distinct, which is why these symptoms can be examined as separate constructs. Additionally, we addressed possible limitations due to the assessment of burnout in more depth in the discussion section.

Changes regarding the Introduction (pp. 4, lines 64-70):

Burnout is characterised by exhaustion, disengagement and reduced self-efficacy^{7,8}. Longitudinal empirical studies suggest that exhaustion is the primary symptom of burnout. In the academic context, sustained study-related stress can lead students to feel emotionally drained and perceive their studies as burdensome. As a coping response, students may begin to disengage from their studies, which, over time, can

undermine their sense of competence and academic efficacy⁹⁻¹¹. Accordingly, exhaustion is considered the initial and central symptom of academic burnout in this study.

Further changes regarding the discussion (pp. 23-24, lines 570-579):

First, we only assessed exhaustion, as the starting point of the burnout process⁹⁻¹¹, but not disengagement or reduced self-efficacy, which are also characteristics of burnout^{7,8}. Furthermore, we assessed exhaustion using a general momentary affect scale to capture subjective experiences of stress, exhaustion and low energy. Although this differs from the usual domain-specific burnout measures, we consider it quite plausible that students' momentary exhaustion levels are to a meaningful degree determined by stress related to university, given that the data was collected during a high-stakes exam period. However, we acknowledge that using a non-specific exhaustion measure may limit the validity and interpretability of our findings, a limitation that should be addressed in future studies.

Parker, P. D., & Salmela-Aro, K. (2011). Developmental processes in school burnout: A comparison of major developmental models. *Learning and Individual Differences*, 21(2), 244–248. <https://doi.org/10.1016/j.lindif.2011.01.005>

Roskam, I., & Mikolajczak, M. (2021). The slippery slope of parental exhaustion: A process model of parental burnout. *Journal of Applied Developmental Psychology*, 77, Article 101354. <https://doi.org/10.1016/j.appdev.2021.101354>

Salmela-Aro, K., Kiuru, N., Leskinen, E., & Nurmi, J.-E. (2009). School Burnout Inventory (SBI): Reliability and validity. *European Journal of Psychological Assessment*, 25(1), 48–57. <https://doi.org/10.1027/1015-5759.25.1.48>

Taris, T. W., Le Blanc, P. M., Schaufeli, W. B., & Schreurs, P. J. G. (2005). Are there causal relationships between the dimensions of the Maslach Burnout Inventory? A review and two longitudinal tests. *Work & Stress*, 19(3), 238–255. <https://doi.org/10.1080/02678370500270453>

7. This is minor, but could you describe how self-esteem instability was measured in the method? It is described in a footnote in Table 1, but probably would fit better under “self-esteem” in the materials, or otherwise in the data analysis strategy so that it is clear when reading the paper in order from start to finish.

Thank you for this helpful suggestion. We agree that the description of the computation of self-esteem instability is more appropriately placed in the main text. We have now added a short explanation to the materials section (pp. 11, lines 270-272):

To quantify self-esteem instability, we calculated the within-person standard deviation of momentary self-esteem ratings across all prompts for each participant. Higher values reflect greater instability in self-esteem over time.

8. The analysis strategy in general seems well thought out, though it is somewhat unusual. I have seen models where (a) variance is partitioned into between vs. within components with person mean centering and (b) lagged models where scores at t predict $t+1$ separately, but I have not seen them combined together in a single model as you have here before. Could you walk me through the rationale for doing the model in this way, and/or direct me to any prior research that analyzed data via this method? Why do both (a) and (b) instead of just one or the other? You cite Wang & Maxwell (2015), but I think they only talk about (a). I suspect this would also involve describing the rationale for your study method/design in Figure 1, which is somewhat more complex than simply measuring all variables at all prompts.

We thank the reviewer for raising this thoughtful point regarding our modeling approach. We appreciate the opportunity to clarify our rationale more explicitly, as this may not have been sufficiently clear in the original manuscript. As correctly noted, Wang and Maxwell (2015) only describe how within- and between-subject variation can be disentangled, which is a critical step regardless of whether one investigates concurrent or lagged associations. Failing to distinguish these levels of variation can lead to conflation of within- and between-subject effects. In turn, the use of lagged predictors allows for examining temporal dynamics of the relationships of interest. The ordering of variables in our analyses was informed by theoretical considerations, as described in the Introduction. Combining the investigation of lagged relationships and separating within- and between-subject effects is well established and common in ecological momentary assessment research. For example, Kirtley et al. (2022) examined how daily fluctuations in positive and negative affect (personal-mean centered) predicted next-day future thinking.

Reference to Kirtley et al. and additional examples:

- Kirtley, O. J., Lafit, G., Vaessen, T., Decoster, J., Derom, C., Gülöksüz, S., De Hert, M., Jacobs, N., Menne-Lothmann, C., Rutten, B. P. F., Thiery, E., Van Os, J., Van Winkel, R., Wichers, M., & Myin-Germeys, I. (2022). The relationship between daily positive future thinking and past-week suicidal ideation in youth: An experience sampling study. *Frontiers in Psychiatry*, 13, 915007. <https://doi.org/10.3389/fpsyt.2022.915007>
- Hoyniak, C. P., Vogel, A. C., Puricelli, A., Luby, J. L., & Whalen, D. J. (2024). Day-to-day bidirectional associations between sleep and emotion states in early childhood: Importance of end-of-day mood for sleep quality. *Sleep Health*, 10(3), 264–271. <https://doi.org/10.1016/j.sleh.2023.12.007>
- Ng, A. S. C., Massar, S. A. A., Bei, B., & Chee, M. W. L. (2023). Assessing ‘readiness’ by tracking fluctuations in daily sleep duration and their effects on daily mood, motivation, and sleepiness. *Sleep Medicine*, 112, 30–38. <https://doi.org/10.1016/j.sleep.2023.09.028>
- Welhaf, M. S., Mata, J., Jaeggi, S. M., Buschkuehl, M., Jonides, J., Gotlib, I. H., & Thompson, R. J. (2024). Mind-wandering in daily life in depressed individuals: An experience sampling study. *Journal of Affective Disorders*, 366, 244–253. <https://doi.org/10.1016/j.jad.2024.08.111>

9. When reporting slopes for various coefficients, consider also reporting some measure of the slope’s variability. Ideally, a 95% confidence interval, but a standard error would also be suitable for this purpose. If you were tight on words, 95% CIs could replace the in-text p-values, especially since most of the p-values are $< .001$.

Thank you for this helpful suggestion. We initially attempted to include random slopes for time-varying predictors, but the majority of these models failed to converge. Therefore, we proceeded with including random intercepts only. We have now added a sentence clarifying this decision in the Statistical Analysis section (see, p. 14, lines 316-318):

Random slopes for time-varying predictors were initially specified, but most models failed to converge. Therefore, only random intercepts were included.

Aside from this, we have added the confidence intervals for the fixed effects.

10. This is optional, but consider figures in the paper (or online supplement) visualizing the interaction effects (e.g., simple slopes plots), as it might be helpful for readers to grasp the overall pattern better.

Thank you for this suggestion! We have now added Johnson-Neyman plots for the moderation analyses (see Figure 3):

11. If I'm following the results on the moderation in lines 318-331 properly, it is a somewhat complicated pattern inasmuch as self-esteem instability attenuates the a-path, but amplifies the b-path. Since the interaction term coefficient is larger for the amplification than for attenuation (i.e., coefficients of 0.09 vs. 0.03), the net effect on the indirect effect is amplification (i.e., the indirect effect is larger with higher self-esteem instability). Am I following that right? If so, this could be made a little clearer in the discussion, as you don't talk about the moderated indirect effect at all there (only the a-path and b-path separately).

Thank you for this thoughtful comment. Your interpretation of the pattern – namely that self-esteem instability attenuates the a-path but amplifies the b-path, with a net amplification of the indirect effect – was indeed correct based on the originally reported results. However, in response to your comment, we re-examined our analysis code and discovered an error in the bootstrapping procedure for testing differences in the indirect effect between individuals with low vs. high self-esteem instability. After correcting this error, the difference of the indirect effect between individuals with low vs. high self-esteem instability is no longer statistically significant. Thus, the opposing

moderation effects on the a- and b-paths appear to cancel each other out. We apologize for this mistake and have since thoroughly reviewed all analysis code to ensure the accuracy of the remaining results. Also, we have added a paragraph to the discussion on the moderation of the indirect effect (see, p. 22, lines 526-537).

Although self-esteem instability moderated both the a-path (i.e., the association between self-esteem on pre-sleep worry) and the b-path (i.e., the association between pre-sleep worry on next-day burnout symptoms) of the mediation model, these moderating effects were functionally different from a statistical view: Self-esteem instability attenuated the a-path and amplified the b-path. Accordingly, there was no significant moderation of the overall indirect effect. This suggests that while self-esteem instability modulated certain components of the mediation process, it does not systematically attenuate or amplified the overall pathway from self-esteem to burnout symptoms via pre-sleep worry. However, as self-esteem instability attenuated the protective effect of higher self-esteem on pre-sleep worry, but amplified the negative impact of pre-sleep worry on burnout, the two moderation effects are similar in that self-esteem instability undermines students' resilience to stress during demanding academic periods in both cases.

12. In the limitations section, you describe that future research should address bidirectional relationships. Why didn't you do that in your study? Is it a limitation of the somewhat unusual measurement schedule for variables?

That is a valid question. As we have already touched upon in point 1, we considered self-esteem to be more of a precursor than an outcome of burnout, at least in theory. The aim of our study was to identify an internal vulnerability factor for burnout on within-person level, with a view to addressing this in primary prevention programmes. As self-esteem and burnout are (highly) intercorrelated with repetitive negative thinking (Kuster et al., 2012, Shigematsu et al., 2024) we expected this to be a mediator of the relationship. However, against the backdrop of bidirectional effects between self-esteem and rumination (e.g. Li et al., 2024) and further theoretical consideration, it may be worthwhile to consider bidirectional effects between all three variables. This could provide valuable insights into chronification processes at the within-person level.

Kuster, F., Orth, U., & Meier, L. L. (2012). Rumination mediates the prospective effect of low self-esteem on depression: a five-wave longitudinal study. *Personality & social psychology bulletin*, 38(6), 747–759.

<https://doi.org/10.1177/0146167212437250>

Li, X., He, Y., Ye, K., & Xu, H. (2024). The Bidirectional Relationship between Rumination and Self-Esteem: Evidence from Longitudinal Tracking and Diary Methods. *Studia Psychologica*, 66(3), 193-206.

<https://doi.org/10.31577/sp.2024.03.900>

Shigematsu, J., Hako, S., Toyokuni, C. *et al.* Moderating effect of self-compassion in the association of automatic thoughts and rumination with burnout among nursing students. *Curr Psychol* **43**, 36306–36314 (2024). <https://doi.org/10.1007/s12144-024-07062-6>

Dear Ms Brueckmann,

Your manuscript titled "Repetitive Negative Thinking Mediating the Relationship Between Self-Esteem and Burnout in an Ecological Momentary Assessment Study" has now been seen by our reviewers, whose comments appear below. In light of their advice I am delighted to say that we are happy, in principle, to publish a suitably revised version in Communications Psychology.

We therefore invite you to revise your paper one last time to address the remaining concerns of our reviewers and a list of editorial requests. At the same time we ask that you edit your manuscript to comply with our format requirements and to maximise the accessibility and therefore the impact of your work.

EDITORIAL REQUESTS:

[...]

Best regards,

Troby Lui, on behalf of

Hannah Hao

Dear Troby Lui, Dear Hannah Hao,

Thank you for your feedback and the opportunity to provide a final revision. We have made according changes.

Kind regards,

the authors

Reviewer #1 (Remarks to the Author):

My concerns with the initial manuscript were related to how self-esteem and burnout were being measured. Both have been addressed well in this revision.

In addition, I wanted more clarification about how the authors were operationally defining rumination and what recommendations there might be. This has also been adequately addressed in the latest revision.

Thank you!

Reviewer #2 (Remarks to the Author):

Overall, the authors did an excellent job addressing my previous comments and revising the manuscript. They also went above and beyond in some of the analytic components by exploring the reverse direction of lagged relationships and adding JN plots for the interaction. I'm also glad you found the coding error towards the end with the interaction prior to publication, and the results are more sensible now that it is fixed. I have only a few remaining comments to share for you to optionally consider and now consider this paper suitable for publication.

Thank you!

5. I definitely understand not wanting to go the route of multiple imputation. This is already a complex study, and it is quite challenging to implement and might result in uncertain benefits in an analysis like this. That said, I have a couple of small follow-ups. Though it is a subtle distinction, lme4 discards incomplete observations, not incomplete participants. The term "listwise deletion" usually implies keeping only those participants with complete data on all observations, which isn't quite accurate (and in fact, would be inferior to what lme4 actually does!). See this stack overflow post, and Ben Bolker's response specifically as he played a pivotal role in developing lme4 and knows way more than me:

<https://stackoverflow.com/questions/78636723/mixed-effects-models-does-lmer-function-really-do-listwise-deletion>

So, all that to say maybe don't say "listwise deletion" as it is somewhat misleading, and describe it more like in this post.

Thank you for this helpful comment, we have adjusted our manuscript accordingly!

Secondly, your missing data analysis on lines 258-262 is a good addition, though it is inconclusive. This means you don't know if the data are (a) missing completely at random, which means your results are unbiased or (b) missing not at random, based on some unmeasured predictors, which could bias the results. Adding a sentence somewhere in the paper (either around here, or in the limitations section) discussing this uncertainty briefly would be welcome, but is not strictly necessary.

We have now added a corresponding statement in the limitation section.

8. You did provide a number of citations to the method of past empirical studies where the combination of group mean centering and lagged variables are used, but I was actually hoping for a statistical reference. That is, I wasn't looking for evidence of what other people have done, but rather, statistical evidence (e.g., from simulation studies) that this approach is desirable. Reviewing all of the studies you cite briefly, it doesn't look like any of them have a statistical reference citing this approach either, so it is possible such a citation does not exist. I did a little bit of research on the subject, I'd like to just leave you with this citation, suggesting that level 1 autoregressive parameters are biased downwards (i.e., too small) when cluster mean centering is used when compared to uncentered predictors, which is the closest analogue I could find to what you did (cluster mean centering with lagged variables, but not autoregressive ones). To be clear: I'm not asking you to re-analyze any data, but rather just sharing this with you in the interest of possibly re-thinking the use of cluster mean centering and lagged variables simultaneously in the future in the interest of collegial information sharing, as I think the properties of models with both procedures combined may be unknown at present:

Hamaker, E. L., & Grasman, R. P. (2015). To center or not to center? Investigating inertia with a multilevel autoregressive model. *Frontiers in psychology*, 5, 1492. <https://doi.org/10.3389/fpsyg.2014.01492>

Thank you for this insightful comment. We have also added a corresponding statement in the limitations section.

Finally, a two typos I noticed in the revisions you might consider fixing prior to publication:
Line 55: "...Higher levels of burn out" should be "burnout"
Line 122, the direct quotation is missing the page number which I think is normally included

Thank you, we have corrected these typos!